# Rapid Synthesis Method of Ag_3_PO_4_ as Reusable Photocatalytically Active Semiconductor

**DOI:** 10.3390/nano13010089

**Published:** 2022-12-24

**Authors:** Zsejke-Réka Tóth, Diána Debreczeni, Tamás Gyulavári, István Székely, Milica Todea, Gábor Kovács, Monica Focșan, Klara Magyari, Lucian Baia, Zsolt Pap, Klara Hernadi

**Affiliations:** 1Department of Applied and Environmental Chemistry, Faculty of Science and Informatics, University of Szeged, Rerrich Béla sqr. 1, 6720 Szeged, Hungary; 2Nanostructured Materials and Bio-Nano-Interfaces Center, Interdisciplinary Research Institute on Bio-Nano-Sciences, Babeș-Bolyai University, Treboniu Laurian str. 42, 400271 Cluj-Napoca, Romania; 3Doctoral School in Physics, Faculty of Physics, Babes-Bolyai University, M. Kogălniceanu 1, 400084 Cluj-Napoca, Romania; 4Faculty of Medicine, Iuliu Hatieganu University of Medicine and Pharmacy, Victor Babeş 8, 400012 Cluj-Napoca, Romania; 5Department of Horticulture, Faculty of Technical and Human Sciences, Sapientia Hungarian University of Transylvania, Aleea Sighișoarei 1C, 530104 Târgu Mureș/Corunca, Romania; 6Nanobiophotonics and Laser Microspectroscopy Center, Interdisciplinary Research Institute on Bio-Nano-Sciences, Treboniu Laurian str. 42, Babes-Bolyai University, 400271 Cluj-Napoca, Romania; 7Faculty of Physics, Babeș-Bolyai University, M. Kogălniceanu str. 1, 400084 Cluj-Napoca, Romania; 8Institute of Research-Development-Innovation in Applied Natural Sciences, Babes-Bolyai University, Fântânele str. 30, 400294 Cluj-Napoca, Romania; 9Institute of Physical Metallurgy, Metal Forming and Nanotechnology, University of Miskolc, Miskolc-Egyetemváros, 3515 Miskolc, Hungary

**Keywords:** silver phosphate, reusability, precipitation method, photoluminescence, photocatalysis

## Abstract

The widespread use of Ag_3_PO_4_ is not surprising when considering its higher photostability compared to other silver-based materials. The present work deals with the facile precipitation method of silver phosphate. The effects of four different phosphate sources (H_3_PO_4_, NaH_2_PO_4_, Na_2_HPO_4_, Na_3_PO_4_·12 H_2_O) and two different initial concentrations (0.1 M and 0.2 M) were investigated. As the basicity of different phosphate sources influences the purity of Ag_3_PO_4_, different products were obtained. Using H_3_PO_4_ did not lead to the formation of Ag_3_PO_4_, while applying NaH_2_PO_4_ resulted in Ag_3_PO_4_ and a low amount of pyrophosphate. The morphological and structural properties of the obtained samples were studied by X-ray diffractometry, diffuse reflectance spectroscopy, scanning electron microscopy, infrared spectroscopy, and X-ray photoelectron spectroscopy. The photocatalytic activity of the materials and the corresponding reaction kinetics were evaluated by the degradation of methyl orange (MO) under visible light. Their stability was investigated by reusability tests, photoluminescence measurements, and the recharacterization after degradation. The effect of as-deposited Ag nanoparticles was also highlighted on the photostability and the reusability of Ag_3_PO_4_. Although the deposited Ag nanoparticles suppressed the formation of holes and reduced the degradation of methyl orange, they did not reduce the performance of the photocatalyst.

## 1. Introduction

Since the discovery and first application attempts of photocatalysis, numerous materials, such as TiO_2_-based semiconductors and composites, other metal oxides, and salts have been investigated regarding their degradation of organic pollutants [1]. However, even the TiO_2_-based ”flagships” (such as P25 [2] or other commercial titania) have severe limitations due to their relatively wide band gap, poor quantum efficiency, and/or fast recombination rate of photoexcited electron─hole pairs.

Silver-based semiconductors are promising candidates among the newer claimants to reach the throne of large-scale industrial applications due to their strong visible light response. One of their main benefits is that they can provide Ag nanoparticles that can be deposited in situ on the surface of catalysts. Depending on the light source applied during photocatalytic processes, deposited nanoparticles can act as charge separators (mostly under UV irradiation) or electron carriers (under visible light irradiation). Many photocatalysts (especially the Ag-based ones) are susceptible to photochemical corrosion, resulting in poor stability [3]. However, deposited nanoparticles can be effectively used to prevent the photocorrosion of semiconductors [4].

The application of Ag_3_PO_4_ as a photocatalytic material was first reported by Yi and coworkers [5]. Their investigation focused on water-splitting and degrading organic contaminants in wastewater by utilizing visible light. Ma et al. [6] focused on the structural peculiarities of Ag_3_PO_4_. They found that it can be characterized by a body-centered cubic structure (where all oxygen atoms are adjacent with three Ag and a P atom), a lattice parameter of 6 Å, and the space group of P4-3n [7,8]. Its band gap energy is relatively low (E_g_ ≈ 2.4 eV), which can be mainly modified by fine-tuning its morpho-structural characteristics. During its synthesis, pH and the appearance of pyro- and polyphosphates are key parameters. It is well known that two phosphoric acid molecules can easily condensate, resulting in these phosphates. Since pH can strongly influence the formation of oxo-acid-based Ag salts, it is not surprising that the type of phosphate precursor can be decisive (as it can also strongly influence pH). Accordingly, several phosphate-containing precursors have already been investigated, focusing on the morphology of the target semiconductor, including NH_4_^+^ [9], Na^+^ [10], and K^+^ [11] based agents. Besides precursors, another decisive parameter is the applied synthesis pathway. Considering this aspect, the main methods applied to synthesize Ag_3_PO_4_ salts have been precipitation [9,10,12], ion-exchange [5,13], hydrothermal [14], ultrasonication [15], and other approaches [16]. It has been found that semiconductors prepared by ion-exchange reactions can yield electron─hole pairs with enhanced lifetime, resulting in higher photocatalytic efficiency [17]. Still, a deeper insight into the electronic structure of Ag_3_PO_4_ is necessary to understand the origin of the increased photocatalytic activity.

Thus, in this work, we focus on three main aspects related to the involvement of the Ag_3_PO_4_ semiconductor in photocatalytic approaches as follows:
The effect of different phosphate sources on the synthesis and photocatalytic activity of Ag_3_PO_4_. For this purpose, H_3_PO_4_, NaH_2_PO_4_, Na_2_HPO_4_, and Na_3_PO_4_ · 12H_2_O were used as phosphate sources. In this part, we propose a mechanism for the formation of Ag_3_PO_4_.Investigation of the stability of Ag_3_PO_4_ via recharacterization and reusability measurements.The effect of Ag nanoparticles on Ag_3_PO_4_, shown through the deposition mechanism of Ag nanoparticles. This aspect was demonstrated by photoluminescence measurements and their corresponding kinetic studies.

## 2. Materials and Methods

### 2.1. Materials

All chemicals were used as received: silver nitrate (AgNO_3_, 99.8%, Penta industry; Prague, Czech Republic), methyl orange (MO, analytical reagent, Alfa Aesar; Tewksbury, MA, USA), Milli-Q (MQ; Budapest, Hungary) water, phosphoric acid (H_3_PO_4_, 85%, VWR Chemicals; Radnor, PA, USA), monosodium phosphate (NaH_2_PO_4_; >99.0%, Spektrum-3D; Debrecen, Hungary), disodium phosphate (Na_2_HPO_4_; >99.0%, Sigma-Aldrich; Schnelldorf, Germany), and trisodium phosphate dodecahydrate (Na_3_PO_4_·12H_2_O; analytical reagent, Sigma-Aldrich, Schnelldorf, Germany).

### 2.2. Methods

#### 2.2.1. Synthesis of Ag_3_PO_4_ Semiconductors

A precipitation method was used [13,18] to synthesize Ag_3_PO_4_ microcrystals. Four phosphate sources (MPO_4_: H_3_PO_4_, NaH_2_PO_4_, Na_2_HPO_4_, Na_3_PO_4_·12 H_2_O) and AgNO_3_ were used. The weight ratio of the precursors was: MPO_4_:AgNO_3_ = 3:2.

In each case, two different initial concentrations of phosphate sources were used (0.2 M and 0.1 M). The aqueous solution of phosphate sources was stirred for 5 min then 1.247 g of AgNO_3_ was added. A yellow suspension formed from the colorless and transparent solution. After another 5 min of stirring, the suspensions were washed with 3 × 45 mL of MQ water (for 10 min at 4400 RPM) and dried overnight at 40 °C.

The sample abbreviations used in the manuscript were conceived as follows: Ag_3_PO_4__source_concentration, where concentration is the initial concentration of the used phosphate precursor (example: Ag_3_PO_4__Na_3_PO_4__0.1M denotes the sample that was prepared using 0.1 M Na_3_PO_4_·12 H_2_O as the phosphate source). The word “*after*” was added to the names to designate samples investigated after photocatalytic tests (example: Ag_3_PO_4__Na_3_PO_4__0.1M_*after*).

#### 2.2.2. Characterization and Instrumentation

The X-ray diffraction (XRD) measurements were performed on a Shimadzu 6000 X-ray diffractometer (Kyoto, Japan) at an accelerating voltage of 40 kV (30 mA), operated with CuKα radiation (λ_CuKα_ = 1.54 Å). The XRD patterns were recorded in 2θ range between 15 and 60°, with scan speed 1°·min^−1^.

Fourier transform infrared spectroscopy (FT-IR) measurements were performed on a JASCO-6200 FT-IR spectrophotometer (Jasco, Tokyo, Japan) in the 4000─400 cm^−1^ wavelength range, with 4 cm^−1^ spectral resolution, using the well-known KBr pellet technique.

The morphology of the samples was identified with a Hitachi S-4700 Type II scanning electron microscope (SEM; Hitachi, Tokyo, Japan) equipped with an Everhart—Thornley detector using an electron beam with an acceleration voltage of 10 kV.

The band structure of the semiconductors was investigated by diffuse reflectance spectroscopy (DRS). The spectra were recorded in the 250–800 nm range with a JASCO-V650 spectrophotometer (equipped with an ILV-724 integration sphere; Jasco, Wien, Austria) using BaSO_4_ as a reference. The band gap energy values were calculated based on the Kubelka-Munk theory [19].

X-ray photoelectron spectroscopy (XPS) measurements were carried out using a Specs Phoibos 150 MCD system (SPECS Surface Nano Analysis GmbH, Berlin, Germany) equipped with Al-Kα source (1486.6 eV) at 14 kV and 20 mA, a hemispherical analyzer, and a charge neutralization device. Care was taken to completely cover the double-sided carbon tapes with the silver—phosphate samples.

Fluorescence measurements were carried out using a Jasco LP-6500 spectrofluorometer (Jasco, Japan; PL) equipped with a Xenon lamp (excitation source) coupled to an epifluorescence accessory (EFA 383 module). Fluorescence spectra were collected with a 1 nm spectral resolution in the 350─600 nm wavelength range using a fixed excitation wavelength of 325 nm. Bandwidths of 1 nm or 10 nm were employed during excitation and emission.

#### 2.2.3. Photocatalytic Activity

The photocatalytic investigation of the samples was carried out in a double-walled photoreactor, where MO (C_MO_ = 125 μM) was the model pollutant. The reactor was surrounded by 6 × 15 W visible light-emitting lamps (λ > 400 nm). The system was kept at a constant temperature (25 °C), and the suspension (C_suspension_ = 1 g∙L^−1^) was continuously stirred and purged by air at constant flow (40 L∙h^−1^). The concentration change of MO was followed with a JASCO-V650 UV-Vis spectrophotometer (UV-Vis; Jasco, Wien, Austria) at λ_max_ = 513 nm (using a 1 mm optical path length quartz cell). The suspension was kept in the dark for 10 min to reach the adsorption–desorption equilibrium. The experiments were conducted for 2 h. Samples were taken every 10 min in the first hour and every 20 min in the second hour. Last, the samples were centrifugated and filtrated before quantitative analysis.

The conversion of MO was calculated by the following equation:H = (100 − (C_120_/C_0_ × 100),
where C_120_ is the concentration of MO after 120 min and C_0_ is the initial concentration of MO.

The reaction order (n) and apparent rate constants (k_1_, k_2_) were calculated to investigate the kinetics of MO degradation. The apparent rate constants (k_1_, k_2_) were determined by plotting the MO concentration vs. the irradiation time, where the slope was considered the apparent rate constants. For k_1_ values, we took the first hour (0–60 min) of the degradation process into account, whereas for k_2_ values, we considered the second (60–120 min) hour.

The adsorption of MO on the samples was also investigated. The photocatalysts (50 mg) and MO (50 mL; C_MO_ = 125 μM) were added to a beaker. The beaker was stirred (500 RPR) and covered with aluminum foil to eliminate all light sources. Samples were taken every 5 min in the first 30 min, every 10 min between 30–60 min, and every 20 min between 1–2 h. Then, they were centrifuged and filtered, and the adsorption of MO was determined with a JASCO-V650 spectrophotometer.

To investigate reusability, we used the same setup for the photoactivity and adsorption measurements, but sampling times were changed to 30, 60, and 120 min. Samples collected between two cycles were washed with distilled water three times and dried at 40 °C for 12 h.

## 3. Results and Discussion

### 3.1. Characterization of Ag_3_PO_4_

Synthesizing Ag_3_PO_4_ by using different MPO_4_ sources is a rather complicated process. MPO_4_ (used as different phosphate sources; Figure 1) has different disproportional rates in aqueous media, which resulted in different pH values (Table 1). Moreover, pH can also indirectly affect the samples’ morphological and structural properties. Using H_3_PO_4_ did not lead to the formation of Ag_3_PO_4_. The lack of Ag_3_PO_4_ can be explained by the acidic environment set by H_3_PO_4_, hindering the precipitation of Ag_3_PO_4_ (Figure 1). Moreover, the presence of H_3_PO_4_ can facilitate the formation of pyrophosphate. Thus, this sample was omitted from all the experimental work presented here.

XRD patterns were recorded to elucidate the effect of the other three phosphate sources on the formation of Ag_3_PO_4_ crystals. Cubic Ag_3_PO_4_ was identified (COD 00-101-0324) in all cases: the reflections of the Ag_3_PO_4_ were located at 2θ° ≈ 21.1°, ≈29.8°, ≈33.3°, ≈36.7°, ≈42.6°, ≈47.9°, ≈52.8°, ≈55.2°, and ≈57.3°, which were assigned to (110), (200), (210), (211), (220), (310), (222), (320), and (321) crystallographic planes (Figure 2a). Additional reflections were also observed in the Ag_3_PO_4__NaH_2_PO_4_ sample series located at 2θ° ≈ 27.0°, ≈30.6° and ≈32.1° (Figure 2a), which could be associated with the typical reflections of Ag_4_P_2_O_7_ [20]. These observations are consistent with the presumed mechanism (Figure 1). It must be emphasized that the pyrophosphate formation is much higher for NaH_2_PO_4_ than for the other two samples. In this case, crystallized pyrophosphate was formed since the materials have higher proton concentrations, and the possibility of condensation is much higher than in other cases. Interestingly, diffraction peaks of other Ag species, such as Ag nanoparticles or Ag_x_O particles, were not found in the XRD patterns (Figure 2a).

FT-IR measurements (Figure 2b) were conducted to investigate the presence of pyrophosphate. The typical bands of asymmetrical vibrations of P─O─P bonds [20] were identified at ≈902 and ≈1116 cm^−1^, confirming the presence of silver pyrophosphate (observed only in the Ag_3_PO_4__NaH_2_PO_4_ sample series; Figure 2b). Additional bands were also observed [17]: at ≈554 cm^−1^ (O=P─O); ≈1007 cm^−1^ (_as_O─P), ≈1389 cm^−1^ (O=P), and H─O─H (≈1655 cm^−1^). Thereby, it can be concluded that the formation of Ag_4_P_2_O_7_ depends on the nature of the applied phosphate source (Figure 2b).

When NaH_2_PO_4_ was used as the phosphate source, the formation of Ag_3_PO_4_ was incomplete, while the formation of Ag_4_P_2_O_7_ was detected (Figure 1). The reason for this could be that the formation of HNO_3_ was not favored. This could be because, during the synthesis, Na^+^ exchange is more favored than that of H^+^. This makes the free formation of NaNO_3_ favorable because the electropositivity of Na^+^ is higher than that of H^+^. Against the Ag_3_PO_4__NaH_2_PO_4_ samples, the Ag_4_P_2_O_7_ was not detected using Na_2_HPO_4_ or Na_3_PO_4_ as a phosphate source.

The morphological properties of the samples were analyzed using SEM. A correlation was found with the XRD measurements (Figure 2a). Two differently shaped and sized particles—*spherical-like* structure with 1.5 μm diameter and *plates* with 0.2 μm height—were obtained in the Ag_3_PO_4__NaH_2_PO_4_ sample series (Figure 3). The particles could not be distinguished based on their elemental composition. This can be attributed to the co-presence of phosphate and pyrophosphate according to XRD patterns (Figure 2a). The samples prepared by using the other two phosphate sources had a much higher monodispersity (Appendix A) and a more defined shape (Figure 3). Their average particle size was ≈0.9 μm regardless of the used concentration. The particle size distribution was lower than for the Na_3_PO_4_ samples series; moreover, lower concentration resulted in smaller particles. The Ag_3_PO_4__Na_2_HPO_4_ sample series contained more polyhedral particles compared to the Ag_3_PO_4__Na_3_PO_4_ sample series.

Diffuse reflectance spectroscopy was used to analyze the optical properties of the materials and to understand the light absorption properties of Ag_3_PO_4__NaH_2_PO_4_ (which contained Ag_4_P_2_O_7_, a stand-alone photocatalyst [20]). As shown in Figure 4a, using different phosphate sources significantly influenced the visible light absorption properties of the samples, whose reflectance intensities decreased in the following order:

Ag_3_PO_4__Na_2_HPO_4_
> Ag_3_PO_4__NaH_2_PO_4_
> Ag_3_PO_4__Na_3_PO_4_. These differences may result in different photocatalytic performances. On the other hand, no specific plasmon resonance band of Ag was detected (which could be found at ≈320 nm [21]). These observations agreement with the XRD results, where the reflection of Ag could not be detected.

The Kubelka-Munk theory was used to calculate the indirect band gap energies of the samples. No significant changes could be observed between them (E_g_ ≈ 2.22–2.34 eV; Table 1). Hence no clear conclusions could be drawn. To understand the relationship between the samples’ light absorption and photocatalytic activity, we analyzed their first derivative spectra (Figure 4b). Still, the λ_max_ values were almost identical (λ_max_ ≈ 495─507 nm; Figure 4). Since Ag_4_P_2_O_7_ can be photoactive as well [20], the derivative DRS of sample Ag_3_PO_4__NaH_2_PO_4_ should result in two specific electron transition peaks: (i) one corresponding to Ag_4_P_2_O_7_ (observed at ≈300 nm [20]) and (ii) one corresponding to Ag_3_PO_4_, (observed at ≈548 nm (2.26 eV [11])).

The lack of the Ag nanoparticles was also demonstrated with high-resolution XPS (Figure 5). Symmetrical peaks were found in the Ag3d spectra (Ag 3d_5/2_ and 3d_3/2_ of Ag_3_PO_4_ corresponding to the peaks 373.67 and 367.67 eV, respectively [22]), which could be associated with Ag^+^ from Ag_3_PO_4_.

### 3.2. Photocatalytic Degradation

MO was employed as a model pollutant to investigate photoactivity. As shown in Figure 6, the photocatalytic activity of the samples can be correlated with the phosphate source used in the synthesis process. Regardless of the concentration of the phosphate source, the order was as follows: Ag_3_PO_4__Na_3_PO_4_
> Ag_3_PO_4__Na_2_HPO_4_
> Ag_3_PO_4__NaH_2_PO_4_ (Figure 6). Because the samples had different absorption properties, similar photocatalytic activation was also expected to differ. According to our assumptions, samples containing Ag_4_P_2_O_7_, a stand-alone photocatalyst, should have higher photocatalytic activity. Moreover, a photojunction between Ag_4_P_2_O_7_ and Ag_3_PO_4_ could also form [23]. Ag_4_P_2_O_7_ did not show any spectral features, that is, its characteristic secondary bands were not found in the DR and first-order derivate spectra (Figure 4). It could inhibit the photocatalyst and absorb electrons, which in turn could not be used in the photocatalytic process. As a result, a photojunction did not form. Similar to the DRS measurements, regardless of the used phosphate source, different photoactivity was observed (Figure 6). Using different phosphate sources resulted in different optical and structural properties, which influenced the degradation of MO. To clarify the degradation of MO, we evaluated the MO conversion values presented in Section 2.2.3. After two hours, 95.94 and 94.53% conversions were obtained for Ag_3_PO_4__Na_3_PO_4__0.2M and Ag_3_PO_4__Na_3_PO_4__0.1M, respectively (Table 1). It could also be noticed that the most photoactive materials (Ag_3_PO_4__Na_3_PO_4_; Figure 6) had the highest degradation yield of MO degradation after one hour. On the other hand, the second-highest MO decolorization was achieved by using Na_2_HPO_4_ as a phosphate source. Based on these findings, we ascertained that the proposed mechanism of Ag_3_PO_4_ formation (Figure 1) is in good agreement with the photocatalytic performance. The more complete the transformation, the higher the photoactivity because when Na_3_PO_4_ was used during the synthesis, no Ag_4_P_2_O_7_ or Ag nanoparticles could be observed.

As a parallel measurement, the adsorption capacity of the two most efficient photocatalysts was also investigated without using any light source (Appendix A). Adsorption did not occur throughout the process. Thus, it could be concluded that the photocatalytic degradation of MO could indeed be performed using these Ag-based materials.

Amornpitoksuk et al. [10] investigate the effect of different phosphates on the photocatalytic degradation of methylene blue. They found that using Na_2_HPO_4_ resulted in the highest photocatalytic activity. They also mentioned that the samples synthesized from Na_3_PO_4_ contained not only Ag_3_PO_4_ but Ag_2_O as well, which inhibited photoactivity. Similarly, in our case, samples that did not contain Ag_2_O had the highest photocatalytic activity. However, it cannot be conclusively declared which phosphate source is the most suitable for achieving high photoactivity. In our case, the samples containing pyrophosphate proved to be less effective. Thus, it can be concluded that synthesizing pure Ag_3_PO_4_ is the best approach since both Ag_2_O and Ag_4_P_2_O_7_ are stand-alone photocatalysts that do not improve the efficiency of Ag_3_PO_4_.

Regardless of the employed phosphate source, after one hour (Figure 6), a change in the photocatalytic reaction was observed during MO removal. A kinetic study of the MO degradation curves was carried out.

Regarding the kinetics of MO degradation, a two-step mechanism was proposed (Table 2). The first step is the zero-order decolorization of azo dye bonds (R_1_─N=N─R_2_), probably by direct hole oxidation (0–60 min) since this is the thermodynamic and electrochemical facile pathway [24]. The second step (k_2_) is the first-order mineralization (60–120 min) of the intermediates (benzenesulfonic acid; 4-hydrazinylaniline; phenyl diazene) by reactive radicals (•OH; O_2_^•─^) [25]. The k_1_ values are in excellent agreement with the correlation coefficients (R^2^) regarding the MO conversion values after 1 h of visible light irradiation. The proposed mechanism can be applied to most samples; however, for Ag_3_PO_4_-Na_3_PO_4_-0.2M and Ag_3_PO_4_-Na_3_PO_4_-0.1M (Table 2), the degradation of intermediates follows second-order kinetics [26]. This could mean that the relatively fast mineralization of intermediates could also occur by direct hole oxidation and not by reactive radicals [27,28].

### 3.3. Stability Investigation of Ag_3_PO_4_

#### 3.3.1. Recharacterization of the Investigated Ag_3_PO_4_

After the photocatalytic tests, we once again investigated the structural and optical parameters of the samples to examine the stability of the photocatalysts. After the photocatalytic activity, a slight modification was observed in the XRD patterns (Figure 7a) because not only the typical Ag_3_PO_4_ reflections (presented in Figure 2a), but a specific reflection of Ag (nano)particles was also observed at 38.1° (COD-00-110-0136). Based on our previous measurements [29,30], a silver-based material might lose photocatalytic activity due to the presence of Ag nanoparticles on the surface. Still, several publications in the literature state that Ag-based materials can be used quasi-unlimited times in (photo)catalytic processes [10]. Besides the deposited Ag nanoparticles, the typical reflection of Ag_4_P_2_O_7_ was also observed after the photocatalytic degradation, which was noticed before the degradation. Moreover, the optical parameters (Figure 7b–c) were also evaluated, and the decreasing intensity of the absorption band of Ag_3_PO_4_ was observed. The typical plasmon resonance band could not be clearly identified. Thus, it can be concluded that the amount of Ag nanoparticles was lower than the detection limit of the device applied. The appearance of Ag_x_O is also possible because their reflection is close to that of Ag, and they do not have plasmon resonance bands in the reflectance spectra.

#### 3.3.2. Reusability of Ag_3_PO_4_ Samples: Recycling Tests and Photoluminescence Measurements

To investigate reusability, we have chosen the Ag_3_PO_4__Na_2_HPO_4__0.2M and Ag_3_PO_4__Na_3_PO_4__0.2M samples because these were the most efficient photocatalysts with a degradation of 89% and 95%, respectively (Table 1). The samples had photocatalytic activity after the second round as well (Figure 8a); however, a decrease in the degradation of MO was observed. The reason for this decrease was investigated by photoluminescence measurements (Figure 8b) to examine the recombination properties of Ag_3_PO_4_. Similar to Mohammadreza Batvandi et al. [31], two of the observed emission bands in our samples were in the violet and blue-cyan wavelength regions. The emission band observed in the violet region corresponds to charge transfer and self-trapping [32]. The blue-cyan wavelength can be assigned to surface oxygen vacancies and defects. After degradation, the intensity of these bands decreased, and in parallel, the band in the violet region increased slightly. The reason for the decreasing bands may be the disappearance of surface oxygen vacancies and defects, which may result from the deposition of Ag nanoparticles on the semiconductor surface. Based on our interpretation, the disappearance of oxygens vacancies and significant disappearances of defects resulted in a decrease in photocatalytic activity.

Considering all the results presented above, we proposed a possible charge transfer mechanism (Figure 9). Before elaborating on the mechanism, we summarized the main conclusions that are indispensable for understanding the mechanism as follows:Besides the Ag_3_PO_4_, only Ag_4_P_2_O_7_ was observed after the synthesis. Typical bands for surface oxygen vacancies and defects were also observed in the PL spectra.Two different pathways were observed using MO as a model pollutant, and the kinetics parameters changed after the first hour.The deposition of Ag nanoparticles was observed after MO degradation, which resulted in the lack of surface oxygen vacancies and defects.

Defects and vacancies are essential for the degradation of MO. The degradation of MO occurs by reactive radicals/h^+^ formed in the valence bands of materials. After irradiation, structural changes were observed in our materials. We assume that Ag nanoparticles (Figure 9) overlap with or replace vacancies and defects (*demonstrated by PL measurements*; Figure 8b). These nanoparticles change the reaction mechanism, which was confirmed by the change in the kinetic parameters (*from 0th→1st and 2nd order reaction;* Table 2). This assumption was confirmed by the typical band observed in the PL spectra, attributed to charge transfer and self-trapping (Figure 8b). This also caused fewer holes to form. Thus, the charge transfer between Ag nanoparticles and Ag_3_PO_4_ occurs by utilizing their localized surface plasmon resonance effect (Figure 9). Thereby, it can be deduced that the formed Ag nanoparticles do not modify the photocatalysts; they only change the mechanism of MO degradation by promoting the transfer of electrons to the semiconductor conduction band. Consequently, the formation of holes is no longer favored. The lack of charge carrier holes also results in a lower photocatalytic efficiency, supported by the MO degradation rate slowing down during the 2-h-long experiment. This is also supported by the lower MO degradation in the second run of the reusability tests. In addition, the formation of Ag NPs could also degrade the Ag_3_PO_4_ and could damage the semiconductor via Ag^+^ ions being reduced into Ag NPs. These are observable by the change in the DR spectra of the samples before (Figure 4a) and after (Figure 7b) the degradation.

Although Ag nanoparticles were formed on the surface, they participated in electron transfer (transferring electrons to the conduction band of the semiconductor) processes, which did not promote the degradation of MO.

## 4. Conclusions

This paper examined the effect of different phosphate sources on the synthesis and photocatalytic activity of Ag_3_PO_4_. We proved that the formation of Ag_4_P_2_O_7_ depends on the nature of the phosphate source. The type of phosphate sources influenced the light absorption properties and photocatalytic activity of the samples. We concluded that Ag_4_P_2_O_7_ inhibits the photocatalytic activity of Ag_3_PO_4_. In addition to other similar publications in the literature, we also investigated the stability and reusability of Ag_3_PO_4_. We concluded that Ag species were formed on the Ag_3_PO_4_, which resulted in a slightly lower methyl orange degradation during the reusability processes. The difference could be attributed to the localized surface plasmon resonance of Ag nanoparticles, promoting the transfer of electrons within the semiconductor and preventing hole formation. This fact was supported by PL measurements. Considering the characterization results obtained before and after the photocatalytic tests, we concluded that Ag_3_PO_4_-based materials could be reliably used for the degradation of MO as they mostly retain their photoactivity during second recycling test.

## Figures and Tables

**Figure 1 nanomaterials-13-00089-f001:**
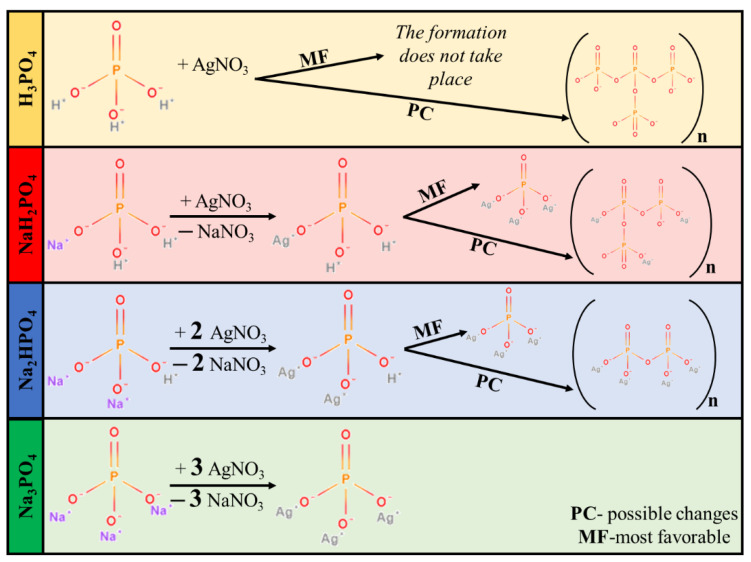
Proposed mechanism of Ag_3_PO_4_ formation.

**Figure 2 nanomaterials-13-00089-f002:**
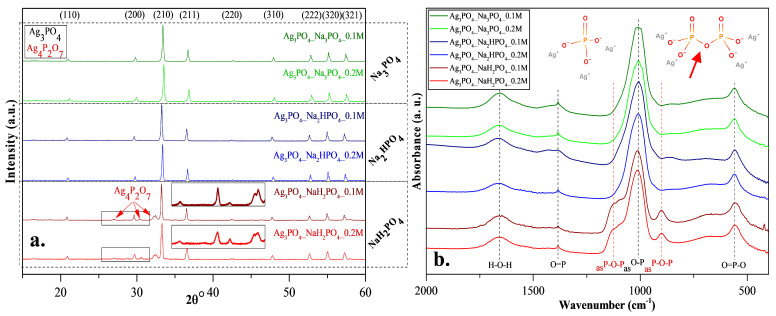
(**a**) X-ray diffraction patterns and (**b**) infrared spectra of Ag_3_PO_4_ samples synthesized by using different types (Na_3_PO_4_, Na_2_HPO_4_ and NaH_2_PO_4_) and concentrations of phosphate sources (C = 0.1 and 0.2 M).

**Figure 3 nanomaterials-13-00089-f003:**
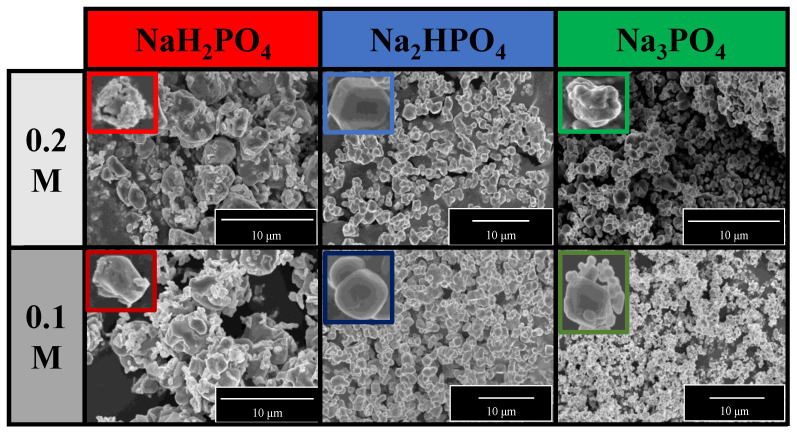
SEM micrographs of Ag_3_PO_4_ samples synthesized by using different types (Na_3_PO_4_, Na_2_HPO_4_, and NaH_2_PO_4_) and concentrations of phosphate sources (C = 0.1 and 0.2 M).

**Figure 4 nanomaterials-13-00089-f004:**
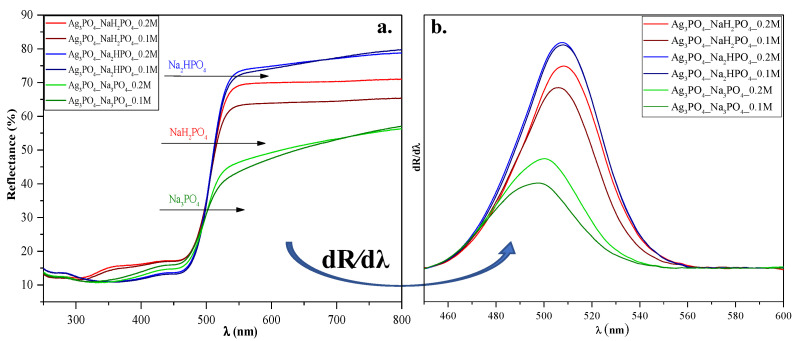
(**a**) Diffuse reflectance spectra and (**b**) their first derivative order of Ag_3_PO_4_ samples synthesized by using different types (Na_3_PO_4_, Na_2_HPO_4_, and NaH_2_PO_4_) and concentrations of phosphate sources (C = 0.1 and 0.2 M).

**Figure 5 nanomaterials-13-00089-f005:**
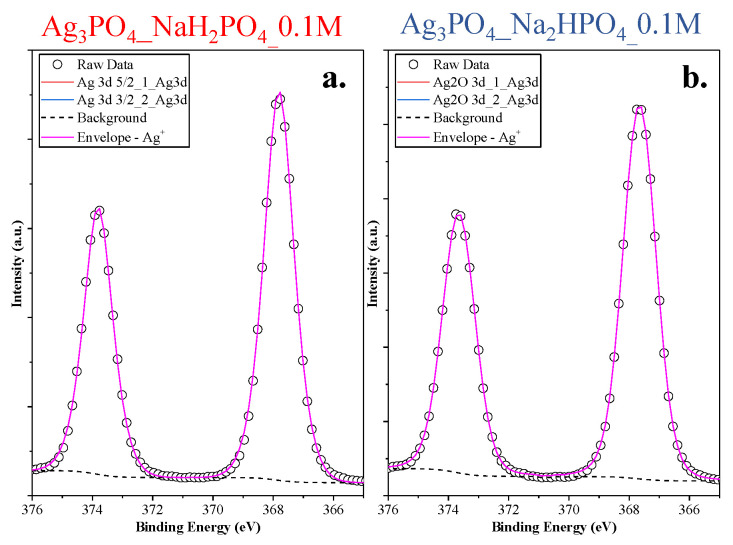
Ag3d XPS spectra of the Ag 3d (**a**) Ag_3_PO_4__NaH_2_PO_4__0.1M and (**b**) Ag_3_PO_4__Na_2_HPO_4__0.1M samples.

**Figure 6 nanomaterials-13-00089-f006:**
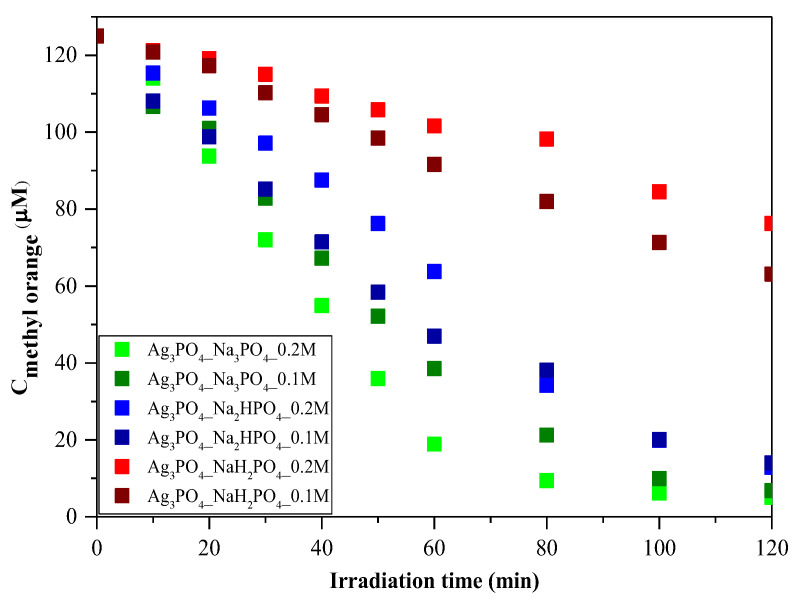
Photocatalytic investigation of Ag_3_PO_4_ samples using C = 125 μM of MO as a model pollutant and visible light (λ > 400 nm) as a light source.

**Figure 7 nanomaterials-13-00089-f007:**
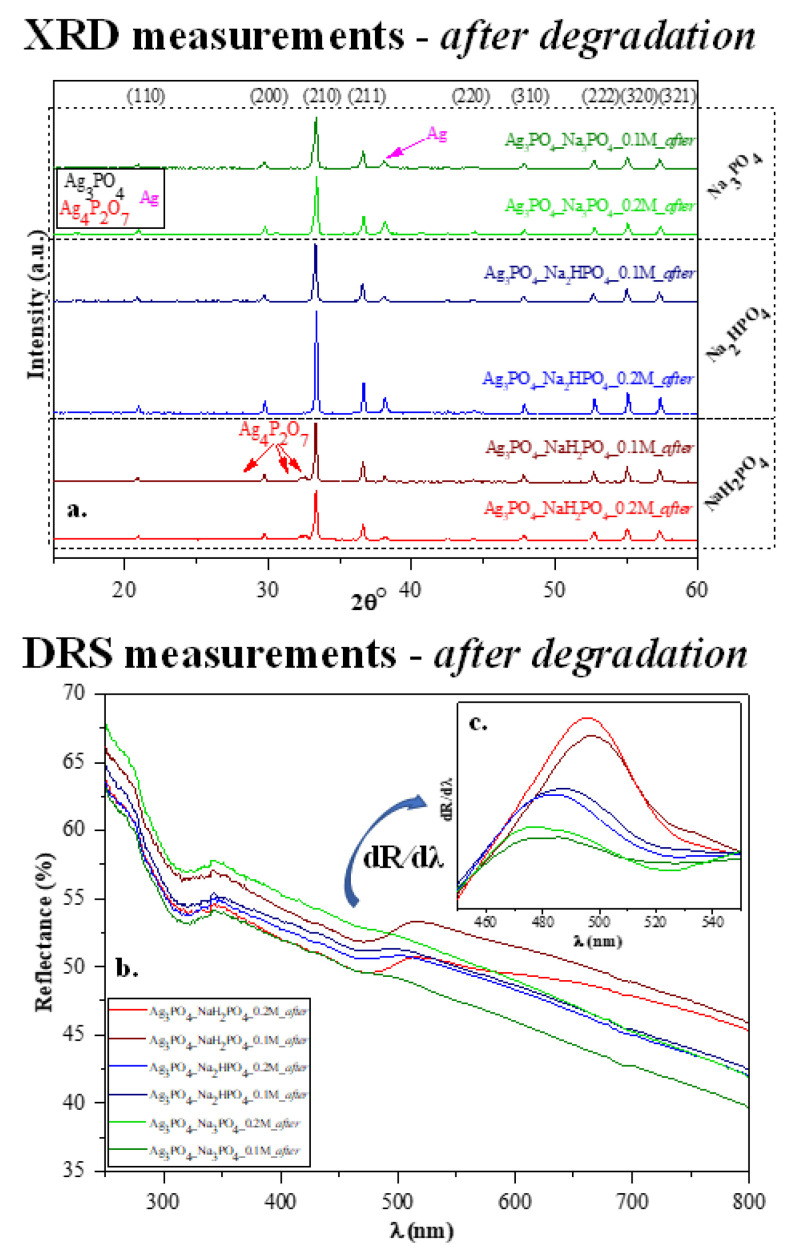
Stability measurements carried out after photocatalytic degradation: (**a**) XRD patterns and (**b**) DR spectra and (**c**) insert figure: their first derivative order. The term “*after*” was used in the sample names to indicate that the results were obtained after MO degradation.

**Figure 8 nanomaterials-13-00089-f008:**
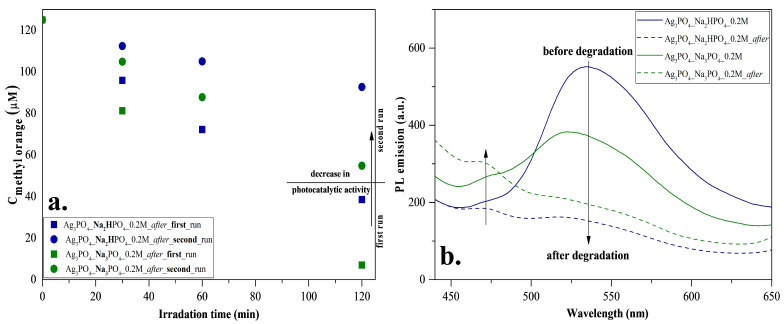
Stability measurements for Ag_3_PO_4__Na_2_HPO_4__0.2M and Ag_3_PO_4__Na_3_PO_4__0.2M: (**a**) reusability test with MO degradation under visible light irradiation and (**b**) photoluminescence spectra at 325 nm excitation.

**Figure 9 nanomaterials-13-00089-f009:**
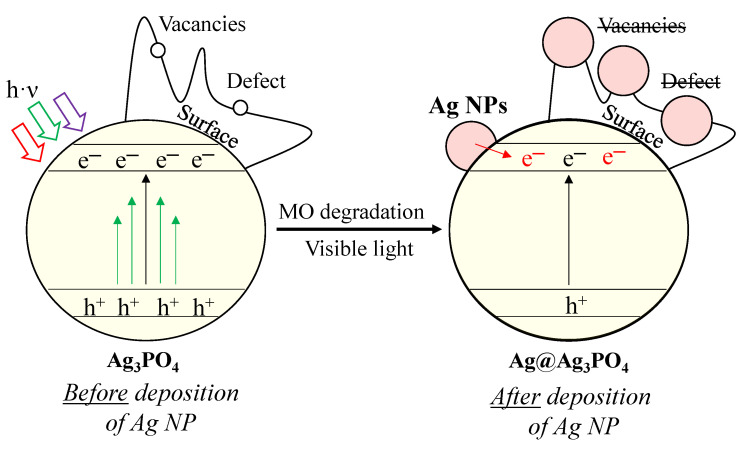
Proposed degradation mechanism of MO with Ag_3_PO_4_ photocatalysts.

**Table 1 nanomaterials-13-00089-t001:** pH, band gap energy, absorption band maxima, average particles sizes, and conversion values of the obtained Ag_3_PO_4_ samples (n.a.—no available).

	pH MPO_4_	Band Gap Energy (eV)	λ_max_ (nm)	đ_SEM_ (μm)	Conversion −2 h-(%)
**Ag_3_PO_4__NaH_2_PO_4__0.2M**	4.22	2.3	507	n.a.	38.98
**Ag_3_PO_4__NaH_2_PO_4__0.1M**	4.27	2.3	504	n.a.	49.53
**Ag_3_PO_4__Na_2_HPO_4__0.2M**	9.14	2.33	506	0.92	89.72
**Ag_3_PO_4__Na_2_HPO_4__0.1M**	9.24	2.34	506	0.97	88.80
**Ag_3_PO_4__Na_3_PO_4__0.2M**	11.46	2.27	498	0.58	95.94
**Ag_3_PO_4__Na_3_PO_4__0.1M**	11.71	2.22	495	0.33	94.53
The colors of the samples are the same as those used in the entire article.					

**Table 2 nanomaterials-13-00089-t002:** Kinetics of methyl orange degradation of the Ag_3_PO_4_ samples.

	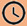 First Hour → 0 Order	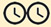 Second Hour → 1 or 2 Order
NaH_2_PO_4_//Na_2_HPO_4_//Na_3_PO_4_→0th Order	NaH_2_PO_4_//Na_2_HPO_4_→1st orderNa_3_PO_4_ → 2nd Order
Sample	Reaction Order (n)	k_1_ (μM∙min^−1^)	R^2^	Reaction Order (n)	k_2_ (s^−1^)	R^2^
**Ag_3_PO_4__NaH_2_PO_4__0.1M**	0	0.3945	0.990	1	0.0063	0.988
**Ag_3_PO_4_-NaH_2_PO_4_-0.2M**	0	0.5626	0.991	1	0.0065	0.998
**Ag_3_PO_4__Na_2_HPO_4__0.1M**	0	1.0027	0.996	1	0.0244	0.996
**Ag_3_PO_4__Na_2_HPO_4__0.2M**	0	1.2889	0.997	1	0.0250	0.974
**Ag_3_PO_4__Na_3_PO_4__0.1M**	0	1.8342	0.996	2	0.0023	0.97
**Ag_3_PO_4__Na_3_PO_4__0.2M**	0	1.4366	0.993	2	0.0025	0.998
**Sample**	**Reaction order (n)**	**k_1_ (** **μ** **M∙min^−1^)**	**R^2^**	**Reaction order (n)**	**k_2_ (s^−1^/** **M^−1^** **)**	**R^2^**

## Data Availability

Not applicable.

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
