# Peer review of "Rapid Synthesis Method of Ag3PO4 as Reusable Photocatalytically Active Semiconductor"

_nanomaterials, 2022, doi:10.3390/nano13010089_

Round 1

Reviewer 1 Report

This work deals with a study on  i) the obtaining and characterization of photocatalytically active Ag3PO4 from four (three) different phosphate sources and ii) the study of the photocatalytic activity of six of the obtained samples in the degradation of MO. This last study includes that of the reaction mechanism and reusability of two out of the catalysts in a second reaction cycle. In my oppinion the work deserves interest for readers working in this research area and could be publishable in Nanomaterials upon revision of a series of points appearing bellow:

* From the discussions of sections 3.1 and 3.2 the photocatalytic activities of the studied samples are compared on the basis of their reflectance values in Figure 4. It is not clear if the experimental DRS values of all catalysts were obtained from samples with the same molar concentrations in all cases (as requires the above mentioned comparison). In oder words, are the units in the ordinate of Figure 4 molar reflectance ones?.

*By the way, the description on the basis of Figure 4, Figure 6 and Figure S2 of the photocatalytic activities, in the first paragraph in section 3.2,( particularly the rows 283-287) is quite unintelligible to the reader. Should be revised.

*In subsection 3.3.2., in addition of the formation of Ag nps which bloks the formation of holes in the photocatalysts as the main cause of the partial loss of catalytic activities, degradation of part Ag3PO4 via Ag+ reduction (which becomes in decreasing of the absorbance in UV DRS spectra of Figure 6 of the fresh and reused catalysts) should be mentioned too.

*Figures 7a and 7b are poor quality. Should be improved  

*According to the results of Figure 8a it seems that the catalysts are stable during the second reaction cycles (linearity of points). In spite of the last the affirmation of the last paragraph on the Conclusions section that "... the catalysts retain their photocatalytic activity during repeated measurements..." is not proper as only two reaction cycles were carried out. Please revise.

In my oppinion this paper should be publishable once the recommendations that I have sent are addressed. 

Author Response

Please, read attached doc file with the answers.

Reviewer 2 Report

Results presented are classical for the investigation of new catalysts.

Authors demonstrate and compare the catalytic activity of different composite photocatalysts in the process of the degradation of methyl orange under visible light.

All synthesis procedures and characterization methods are described quite fully. The conclusions are justified.

Nevertheless, a few questions remain.

1. Why author use “The weight ratio of the precursors was: MPO4: AgNO3=3:2” (line 107)  for the preparation of all samples?

2. What experimental facts allow to conclude “ only the first 60 minutes of the photocatalytic process was considered because the formation of intermediates in this interval is probably negligible” (Line 162)?

Author Response

(The authors gave the same response as above.)
